differential equations/graph theory/complexity

cooperative systems, stability analysis, diakoptics, linear systems, population dynamics

**Author for correspondence:**
Philip Greulich
e-mail: p.s.greulich@soton.ac.uk

# Stability and steady state of complex cooperative systems: a diakoptic approach

Philip Greulich[1,2], Ben D. MacArthur[1,2,3],
Cristina Parigini[1,2] and Rubén J. Sánchez-García[1,2]

[1]School of Mathematical Sciences, [2]Institute for Life Sciences, and [3]Centre for Human Development, Stem Cells and Regeneration, University of Southampton, Southampton SO17 1BJ, UK

PG, 0000-0001-5247-6738; BDM, 0000-0002-5396-9750;
RJS-G, 0000-0001-6479-3028

Cooperative dynamics are common in ecology and population dynamics. However, their commonly high degree of complexity with a large number of coupled degrees of freedom renders them difficult to analyse. Here, we present a graph-theoretical criterion, via a diakoptic approach (divide-and-conquer) to determine a cooperative system's stability by decomposing the system's dependence graph into its strongly connected components (SCCs). In particular, we show that a linear cooperative system is Lyapunov stable if the SCCs of the associated dependence graph all have non-positive dominant eigenvalues, and if no SCCs which have dominant eigenvalue zero are connected by a path.

## 1. Introduction

Cooperative systems are a wide class of dynamical systems characterized by a non-negative dependence between components [1]. Common examples are (bio-)chemical reaction networks with mutually activating interactions and compartmental dynamics, where a conserved quantity transits between different compartments or states [2,3]. However, cooperative systems also include non-conserved replicator dynamics, such as (multi-species) population dynamics, where a population of replicators transits between different states/compartments. Examples for the latter are organisms which transit through life cycles or tissue cells (e.g. stem cells) which proliferate, switch between different phenotypes [4] and differentiate during biological development and in renewing tissues. If one considers the dynamics of sub-populations embedded in a larger population, then the equations describing the system are linear: while a population as a whole may be subject to a nonlinear feedback (for example, by a finite carrying capacity), smaller embedded sub-populations compete neutrally with each other without affecting the population as a whole. This renders the dynamics linear.

In this article, we find conditions for the stability of linear cooperative systems, based on graphical criteria of the underlying dependence graph. In an ecological or biological context, stability of populations is required to maintain ecological equilibrium (population of individuals) or a functional biological tissue (population of tissue cells). In particular, instability of a tissue cell population may lead to cancer, thus the study of a cell population's stability is of high biomedical importance. However, the commonly applied property of asymptotic stability is not viable for linear systems in a biological context, since the only asymptotically stable state is extinction. In these contexts, it is therefore more appropriate to study *marginally stable steady states*, a form of Lyapunov stability.

While general criteria for a cooperative system's stability are well established [5], real-world systems can be very complex, with a large number of variables and complex interactions, in which case their analysis is a highly challenging endeavour. Topological features of trajectories, such as compactness, can theoretically be used to determine stability [6,7], but are in practice difficult to apply without explicitly solving the underlying differential equation.

To simplify the analysis of a system, it is useful to represent it as a directed graph in which dependent variables $x_i(t)$, $i \in \mathbb{N}$, are nodes, and links denote dependence relations between those variables. The Jacobian matrix $J = [(\partial \dot{x}_i / \partial x_j)]$ of such a system can be interpreted as an adjacency matrix of an underlying graph representing the mutual dependence of components. Cooperative systems are then defined by non-negativity of Jacobian's off-diagonal entries, which corresponds to positive-only weights of links in the network. The corresponding Jacobian matrix is a Metzler matrix and thus methods based on non-negative matrices (and the Perron–Fobenius theorem) can be applied to study them [8–10].

A paradigm to study complex systems is the *diakoptic* view (divide-and-conquer) [11]: a large interacting system is decomposed into suitable small subsystems, which are studied in isolation, a task which is usually easier to perform. Then a synthesis of subsystems yields the features of the whole system. The analogy between dynamical systems and graphs may be used to apply graph-theoretical tools to perform such a diakoptic decomposition of the system (i.e. graph) into smaller subsystems which can significantly simplify the analysis of stability features of the respective system.

A diakoptic approach, based on the decomposition of the underlying graph into its *strongly connected components* (*SCCs*) has been used to determine asymptotic stability of cooperative systems [12]. An SCC is a subset of nodes which are all mutually reachable by directed paths. Simply speaking, a system is asymptotically stable if, and only if, all its SCCs, when decoupled from each other, are asymptotically stable. This holds, since the eigenvalues of the system's adjacency matrix are the union of the SCCs' eigenvalues [9,12], which can also easily be checked by evaluating row sums of the dynamical matrix [13]. However, this criterion cannot be straightforwardly generalized to determine marginal stability; conclusions about marginal stability versus instability can in general not be drawn just by considering SCCs in isolation and a system may be unstable even if there is no unstable individual SCC. Only for linear compartmental systems—i.e. cooperative systems that feature a conserved quantity—the existence of a marginally state can be found through analysis of SCCs in isolation: if there is at least one singular SCC, a so-called *trap*, then a non-trivial marginally stable steady state exists [14,15]. However, this criterion cannot be applied to linear cooperative systems in general, when dynamics are not conserved.

Here, we introduce a diakoptic approach to determine the stability of general linear cooperative systems, which is also applicable for non-conserved systems, and allows us to identify conditions for marginal stability. This approach is based on graphical criteria of the underlying dependence graph when decomposed for its SCCs. The stability can then be inferred from (i) the spectrum of the Jacobian matrices of isolated SCCs, and (ii) the hierarchical arrangement of the SCCs. Our main result is theorem 2.9 (illustrated in figure 2) which states that for (marginal) stability to prevail, no SCC may have positive eigenvalues, and any otherwise singular SCCs may not stand in any hierarchical relation to each other, i.e. there may be no (directed) path connecting them. This reflects the principle that the larger and more connected complex systems are, the more likely they are to become unstable [16].

## 2. Results

We consider a generic cooperative linear dynamical system of a positive quantity (mass) $m$ on a directed weighted graph with $n$ nodes, whereby we denote $m_i = m_i(t)$ as the mass on node $i = 1, \ldots, n$ at time $t$. The state vector of the system is $\mathbf{m} = (m_1, m_2, \ldots, m_n)^T$ and the system is written as

$$\frac{\mathrm{d}}{\mathrm{d}t}\mathbf{m}(t) = A\,\mathbf{m}(t),\tag{2.1}$$

for a $n \times n$ real square matrix

$$A = [a_{ij}], \quad \text{with } a_{ij} \geq 0 \quad \text{for } i \neq j. \tag{2.2}$$

The condition $a_{ij} \geq 0$ for $i \neq j$, defines the system as *cooperative*, since $A$ is the Jacobian matrix of system (2.1). We note that the system is not necessarily conserved, i.e. the 'mass' could replicate, such as a population of biological individuals, cells or viruses.

We consider the underlying directed weighted graph $G(A)$ with transposed adjacency matrix $A$, that is, the graph with $n$ nodes and a link from $j$ to $i$, weighted by $a_{ij}$, only if $a_{ij} \neq 0$. This is a finite simple graph with positively weighted edges and arbitrarily weighted self-loops. We wish to relate the stability of the fixed points of (2.1) to the network structure of $G(A)$.

Since $A$ is the Jacobian of (2.1), the stability of a fixed point $\mathbf{m}^*$, defined by $A\,\mathbf{m}^* = 0$ (that is, a 0-eigenvector of $A$), is determined by the spectral properties of $A$. For the system to be asymptotically stable, all the real parts of the eigenvalues of $A$ must be negative. In this case, however, $\det(A) \neq 0$ and the only fixed point $A\,\mathbf{m}^* = 0$ is trivial, $\mathbf{m}^* = 0$. As we are interested in non-trivial solutions, we focus instead on Lyapunov stable fixed points which are at least marginally stable (also called *semi-stable* [5]). This is the case if the eigenvalue of $A$ with largest real part is zero and its geometric multiplicity is equal to its algebraic multiplicity [17]. Our main result is a necessary and sufficient condition on the structure of the graph $G(A)$, for the dynamical system to have non-trivial, marginally stable, non-negative solutions. Note that we call a vector $\mathbf{m}$ (or, similarly, a matrix) *non-negative*, written $\mathbf{m} \geq 0$, if all entries are real and non-negative, and *positive*, written $\mathbf{m} > 0$, if all entries are real and positive.

First, we decompose $G(A)$ into its strongly connected components, as follows. A (sub-)graph is *strongly connected* if for any pair of nodes $i$ and $j$ in the graph there is a directed path from $i$ to $j$ and a directed path from $j$ to $i$, that is, every pair of nodes is mutually reachable. Every directed graph can be partitioned into maximal strongly connected subgraphs, the graph's *strongly connected components* (SCCs). The SCCs of a directed graph $G$ form another graph called the *condensation* of $G$: in it, each node represents an SCC, and if two SCCs in $G$ are connected by at least one link, then the condensation possesses a link between them, in the same direction as in $G$ (figure 1). The condensation of a directed graph is always a directed acyclic graph and, hence, its nodes (the SCCs of $G$) admit a *topological ordering* [18]: an ordering $B_1, B_2, \ldots, B_h$ (from now on, we will identify the $k$th connected component of $G$ with its adjacency matrix $B_k$) such that if there is a link from $B_i$ to $B_j$ then $i \leq j$ (see figure 1 for an example). We can extend the ordering to the nodes of $G$ so that node $u \in B_i$ appears before node $v \in B_j$ whenever $i \leq j$. With respect to this re-ordering and re-labelling of the nodes of $G$, the adjacency matrix $A$ of $G(A)$ becomes a lower triangular matrix

$$A = \begin{pmatrix} B_1 & 0 & 0 & 0 & \cdots \\ C_{21} & B_2 & 0 & 0 & \cdots \\ C_{31} & C_{32} & B_3 & 0 & \cdots \\ \vdots & \vdots & \vdots & \ddots & 0 \\ \cdots & \cdots & \cdots & \cdots & B_h \end{pmatrix}, \tag{2.3}$$

where $h$ is the number of SCCs of $G(A)$, $B_k$ is the adjacency matrix of the $k$th SCC ($1 \leq k \leq h$), and $C_{kl}$ encodes the connectivity from $B_l$ to $B_k$. This is sometimes called the *normal form of a reducible matrix* [19]. If there exist a path from $k$ to $l$ (thus $k \leq l$), we call $B_k$ *upstream* of $B_l$, and $B_l$ is *downstream* of $B_k$. If $B_k$ is connected by a single (directed) link to $B_l$ then we also call $B_k$ *immediately upstream* of $B_l$, and $B_l$ *immediately downstream* of $B_k$. From now on, we will implicitly assume a topological ordering and notation as above.

Since $A$, written in the form (2.3), is a lower triangular block matrix, the characteristic polynomial of $A$, $p_A(\lambda) = \det(\lambda I - A)$, is the product of the characteristic polynomials of the $B_k$'s

$$p_A(\lambda) = p_{B_1}(\lambda) \cdot \cdots \cdot p_{B_h}(\lambda). \tag{2.4}$$

Thus, the spectrum of $A$—seen as a multiset—is the union of the spectra of the $B_k$'s, and the algebraic multiplicity of the eigenvalues is preserved.

Since all off-diagonal elements of $A$ (and hence of each $B_k$) are non-negative, and each $B_k$ is the adjacency matrix of a strongly connected graph, the matrices $B_k$ are irreducible Metzler matrices, for which the Perron–Frobenius theorem applies, to the shifted eigenvalues [20]. Therefore, each matrix $B_k$ has a real eigenvalue $\mu_k$ with (strictly) largest real part, which is simple and has a positive eigenvector. We call $\mu_k$ the *dominant eigenvalue* of the matrix $B_k$.

We now introduce some further terminology. We call each SCC, and equivalently its adjacency matrix $B_k$, a *block* of the system (we use the term 'block' and the notation $B_k$ for both the matrix and its graph). We call a block *critical* if its dominant eigenvalue $\mu_k = 0$, *sub-critical* if $\mu_k < 0$ and *super-critical* if $\mu_k > 0$. Correspondingly,

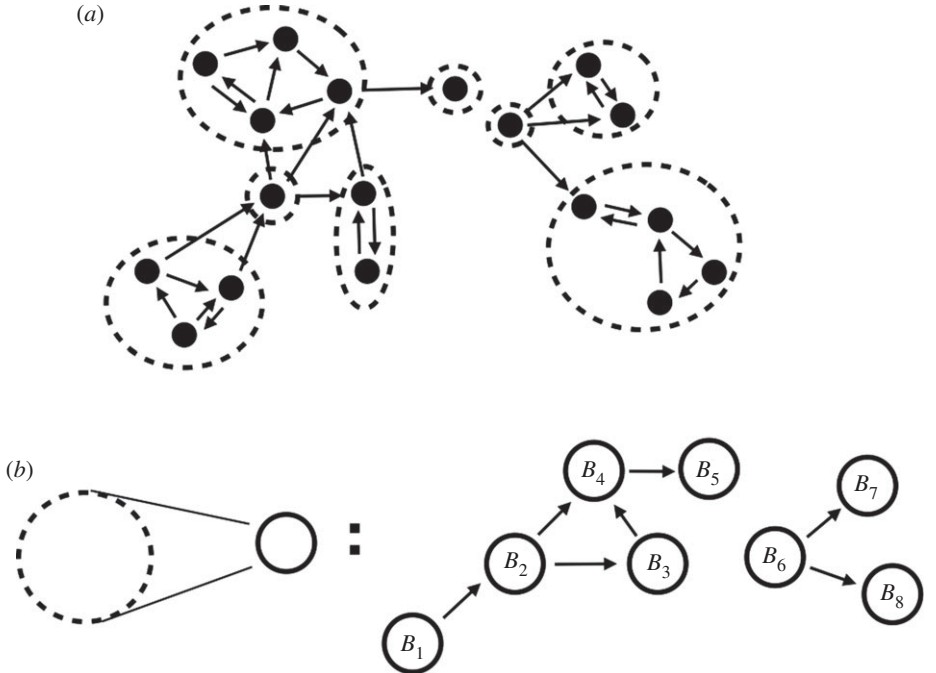

**Figure 1.** Decomposition of a directed graph into SCCs and its condensation graph. (*a*) A directed graph (black dots represent nodes, and arrows directed links) and its SCCs (dashed circles). Note that every node belongs to an SCC, and that an SCC can be a single node. (*b*) Condensation of the directed graph: black circles represent the SCCs and arrows whenever two SCCs are connected via *at least* one link (in the direction shown). The condensation of a graph is always a directed acyclic graph and hence admits a topological ordering, shown here as $B_1$ to $B_8$.

we define the index subsets $I_c = \{k \in 1, \ldots, h \mid B_k \text{ critical}\}$, $I_s = \{k \in 1, \ldots, h \mid B_k \text{ sub-critical}\}$, and $I_{sp} = \{k \in 1, \ldots, h \mid B_k \text{ super-critical}\}$. The first things we note are (e.g. [12])

**Lemma 2.1.** *If at least one block $B_k$ of $A$ is super-critical, then the system* (2.1) *is unstable.*

**Lemma 2.2.** *The system* (2.1) *is asymptotically stable if and only if all blocks $B_k$ of $A$ are sub-critical.*

These lemmas follow immediately from the fact that a system is unstable if at least one real part of an eigenvalue of $A$ is positive, and it is asymptotically stable if and only if all real parts of eigenvalues are negative, together with the property that the spectrum of $A$ is the multi-set union of spectra of the $B_k$ (note, however, that the 'if and only if' statement only holds for lemma 2.2) [12]. In the situation of lemma 2.2, observe that $\det(A) \neq 0$ and hence $\mathbf{m}^* = 0$ is the only fixed point of the system.

Lemmas 2.1 and 2.2 cover all cases where any super-critical blocks exist, or only sub-critical ones. In these cases, the system is either unstable, or has only a trivial (zero) fixed point. In the following, we will consider only the remaining cases when no super-critical blocks exist, but there is at least one critical block, and investigate the existence of non-trivial, non-negative (so that each node supports a non-negative fraction of the 'mass') marginally stable fixed points.

If no super-critical, and at least one critical, block exists, the dominant eigenvalue of $A$ is zero and, according to the Perron–Frobenius theorem, there exist non-trivial eigenvectors $\mathbf{m}^*$ for the eigenvalue zero. It is assured that all such $\mathbf{m}^*$ are equilibrium points of system (2.1); however, to be a (Lyapunov) stable equilibrium it is required that the algebraic multiplicity of eigenvalue zero is equal to its geometric one, or equivalently, equal to the dimension to the nullspace of $A$. We will approach the latter question by explicitly constructing such equilibrium sets.

Let us first write the equilibrium condition of the dynamical system (2.1), using (2.3), as

$$\begin{pmatrix} B_1 & 0 & 0 & 0 & \cdots \\ C_{21} & B_2 & 0 & 0 & \cdots \\ C_{31} & C_{32} & B_3 & 0 & \cdots \\ \vdots & \vdots & \vdots & \ddots & 0 \\ \cdots & \cdots & \cdots & \cdots & B_h \end{pmatrix} \begin{pmatrix} \mathbf{m}_1^* \\ \mathbf{m}_2^* \\ \mathbf{m}_3^* \\ \vdots \\ \mathbf{m}_h^* \end{pmatrix} = 0, \tag{2.5}$$

i.e. the equilibrium vector $\mathbf{m}^*$ is decomposed in the projections $\mathbf{m}_k^*$ on the sub-space of $B_k$, in the form $\mathbf{m}^* = (\mathbf{m_1}, \mathbf{m_2}, \ldots, \mathbf{m_n})^T$. For simplicity, we call $\mathbf{m}_k^*$ the *steady state on* $B_k$. We further call a block $B_k$ *trivial* if $\mathbf{m}_k^* = 0$ for all non-negative marginally stable fixed points $\mathbf{m}^*$ of system (2.5), and *non-trivial* otherwise.[1] In other words, a trivial block is one that does not support any positive fraction of the 'mass' for any non-negative fixed point.

Our first result is a formula for the steady states $\mathbf{m}_k^*$ on sub-critical blocks $B_k$. Let us consider the $k$th row of (2.5),

$$\sum_{l<k} C_{kl} \, \mathbf{m}_l^* + B_k \mathbf{m}_k^* = 0, \tag{2.6}$$

where $B_k$ is a sub-critical block. Since all eigenvalue real parts of $B_k$ are negative, $\det(B_k) \neq 0$ and thus $B_k$ is invertible, so that we obtain a recursive formula for the steady state:

$$\mathbf{m}_k^* = -B_k^{-1} \left[ \sum_{l<k} C_{kl} \, \mathbf{m}_l^* \right]. \tag{2.7}$$

Let $I_f \subseteq I_c$ denote the indices of the critical SCCs for which there are no other critical SCCs downstream. Let us call them *final critical blocks*. With this terminology, we have, from the recursion relation above, the following:

**Theorem 2.3.** *If $B_k$ is a sub-critical block of A and equation (2.5) holds, then $\mathbf{m}_k^*$, the steady state on $B_k$, is uniquely determined by the final critical blocks upstream of $B_k$, namely*

$$\mathbf{m}_k^* = -B_k^{-1} \left[ \sum_{l \in I_f} P_{kl} \, \mathbf{m}_l^* \right], \tag{2.8}$$

*where*

$$P_{kl} = \sum_{(l_1, l_2, \ldots, l_n) \in \mathcal{P}_{kl}} (-1)^{n-1} C_{kl_1} B_{l_1}^{-1} C_{l_1 l_2} B_{l_2}^{-1} \cdots C_{l_n l}, \tag{2.9}$$

*and $\mathcal{P}_{kl}$ is the set of all paths from $B_l$ ($l \in I_f$) to $B_k$, written as a sequence of nodes $(l_1, l_2, \ldots, l_n)$, where n is the length of the path.*

This follows directly if we apply the relation equation (2.7) recursively to all steady states $\mathbf{m}_k^*$ of sub-critical blocks $B_k$ on the right-hand side of equation (2.7), using that when propagating upstream, no critical block can be encountered before a final critical block is encountered.

Theorem 2.3 assures that the steady state on any sub-critical block is uniquely defined by the steady states on all critical blocks upstream of the former. Furthermore, we can conclude:

**Corollary 2.4.** *If $B_k$ is a sub-critical block of A, then $B_k$ is trivial if and only if all $B_l$ immediately upstream of $B_k$ are trivial.*

*Proof.* From equation (2.7), it directly follows that if all $B_l$ immediately upstream are trivial ($\mathbf{m}_l^* = 0$), then $B_k$ is trivial ($\mathbf{m}_k^* = 0$). Now let us consider the case that at least one $B_l$ immediately upstream has $\mathbf{m}_l^* \neq 0$. We first note that since $B_k$ is a Metzler matrix with $\det(B_k) \neq 0$, $-B_k$ is a non-singular M-matrix, and its inverse is a positive matrix (shown in [21]). Thus, $\mathbf{m}_k^*$ is positive if at least one $\mathbf{m}_l^* \neq 0$ (recall that $\mathbf{m}^*$ and the $C_{kl}$'s are non-negative). Therefore, it follows: if $B_k$ is trivial, i.e. $\mathbf{m}_k^* = 0$, then for all immediately upstream $B_l$, $\mathbf{m}_l = 0$, and hence $B_l$ is trivial. ∎

Now we make a topological characterization of the trivial blocks.

**Theorem 2.5.** *A block is trivial if and only if*

(i) *it is upstream of a critical block, or*
(ii) *it is a sub-critical block which is not downstream of a critical block.*

Thereby all trivial blocks can be easily identified by inspecting the condensed graph and its critical blocks (figure 1).

---

[1]Note that $\mathbf{m}_k^*$ is the $k$th sub-space component of the global steady state $\mathbf{m}^*$ of $A$, but not necessarily the steady state of the isolated subsystem of $B_k$.

*Proof.* To prove theorem 2.5, consider an equilibrium point $\mathbf{m}^*$. Then equation (2.5) holds, and in particular its $k$th row equation (2.6). Now $B_k$ is critical and hence has an eigenvalue zero ($\mu_k = 0$ by definition), $\det(B_k) = 0$ and thus $B_k$ is not invertible. Let us multiply both sides of equation (2.6) with the matrix exponential $e^{B_k t} = \sum_{n=1}^{\infty} \frac{(B_k t)^n}{n!}$ to yield

$$e^{B_k t} B_k \mathbf{m}_k^* = B_k [e^{B_k t} \mathbf{m}_k^*] = -e^{B_k t} \left[ \sum_{l<k} C_{kl} \mathbf{m}_l^* \right], \tag{2.10}$$

where we used that a square matrix $M$ commutes with its exponential, $e^M M = M e^M$. In general, $e^{Mt} \mathbf{x}$ is a solution of the linear ODE $\dot{\mathbf{x}} = M\mathbf{x}$ and thus converges to a linear combination of dominant eigenvectors (eigenvectors of the dominant eigenvalues) of $M$. Since $B_k$ is critical, the corresponding dominant eigenvalue is zero and thus $B_k [e^{B_k t} \mathbf{m}_k^*] \to 0$ for $t \to \infty$. This means that $-e^{B_k t} [\sum_{l<k} C_{kl} \mathbf{m}_l^*] = 0$ for $t \to \infty$. Let us call $L = \lim_{t\to\infty} e^{B_k t}$ and $\mathbf{v} = \sum_{l<k} C_{kl} \mathbf{m}_l^*$. We then have $L\mathbf{v} = 0$ and we want to show that $\mathbf{v} = 0$. This is not true for a general vector $\mathbf{v}$ unless $L$ is invertible, but will hold for non-negative eigenvectors such as $\mathbf{v}$. In fact, the matrix $L$ is not invertible in general as all its eigenvalues are zero, except a simple eigenvalue 1 with a positive (left) eigenvector $\mathbf{u}$ (see lemma 2.7 below). In this case, $\mathbf{u}L = \mathbf{u}$ and $\mathbf{u}\mathbf{v} = \mathbf{u}L\mathbf{v} = 0$, a contradiction, since $\mathbf{u}$ is positive and $\mathbf{v}$ is non-negative, unless $\mathbf{v} = 0$.

All in all, we conclude that $\mathbf{v} = \sum_{l<k} C_{kl} \mathbf{m}_l^* = 0$. Note that all entries of the matrices $C_{kl}$ and of the vector $\mathbf{m}_l^*$ are non-negative, so this can only be the case if, for all $l < k$, $C_{kl} = 0$ or $\mathbf{m}_l^* = 0$. Since, for all immediately upstream blocks, we have $C_{kl} \neq 0$, it follows that

**Lemma 2.6.** *All blocks immediately upstream of a critical block are trivial.*

Crucially, from theorem 2.3 and lemma 2.1, it follows that all blocks $B_m$ immediately upstream of any trivial block $B_l$ are trivial (either $B_l$ is critical, or sub-critical and trivial). By applying this argument recursively to lemma 2.1, the first part of theorem 2.5 follows. The second part is an immediate consequence of corollary 2.4. ∎

To complete the proof above, we state and prove the following.

**Lemma 2.7.** *Let $B$ be an irreducible Metzler matrix with shifted Perron–Frobenius eigenvalue 0 and positive left eigenvector $\mathbf{u}$. Then the limit matrix $L = \lim_{t\to\infty} e^{tB}$ exists and satisfies $\mathbf{u}L = \mathbf{u}$.*

*Proof.* Define $f_t(x) = e^{tx}$ for $t > 0$ and write $B$ in Jordan normal form as $B = PJP^{-1}$. By definition [22], the matrix function $f_t(B) = e^{tB}$ equals $Pf(J)P^{-1}$, where $f(J)$ is the block diagonal matrix obtained by applying $f$ to each diagonal Jordan block, $J_i$, of $J$ as

$$J_i = \begin{pmatrix} \lambda_i & 1 & & \\ & \lambda_i & \ddots & \\ & & \ddots & 1 \\ & & & \lambda_i \end{pmatrix} \quad \text{then } f_t(J_i) = \begin{pmatrix} f(\lambda_i) & f_{t'}(\lambda_i) & \cdots & \frac{f_t^{(m_i-1)}(\lambda_i)}{(m_i-1)!} \\ & f_t(\lambda_i) & \ddots & \vdots \\ & & \ddots & f_{t'}(\lambda_i) \\ & & & f_t(\lambda_i) \end{pmatrix}.$$

The eigenvalue of $B$ with largest real part is 0 (dominant eigenvalue), hence $\lim_{t\to\infty} f_t^{(k)}(\lambda_i) = \lim_{t\to\infty} t^k e^{t\lambda_i} = 0$, if $\lambda_i \neq 0$, and 1 otherwise, for all $k \geq 0$. All in all, $L = PMP^{-1}$ where $M$ is the zero matrix except a single 1 in the diagonal. Its eigenvectors (the columns of $P$) are the same as the (generalized) eigenvectors of $B = PJP^{-1}$. Hence the left 0-eigenvector $\mathbf{u}$ of $B$ becomes a left 1-eigenvector of $L$, $\mathbf{u}L = \mathbf{u}$. ∎

We can also easily conclude from theorem 2.5 and equation (2.10):

**Corollary 2.8.** *The steady states $\mathbf{m}_k^*$ on a non-trivial critical block $B_k$ (called a free block) is the one-dimensional family of dominant eigenvectors (of eigenvalue zero) of $B_k$. We can write these as $\alpha_k \phi_k$ where $\alpha_k \in \mathbb{R}$ is a free parameter, and $\phi_k$ is a (normalized) dominant eigenvector of $B_k$.*

Theorems 2.3 and 2.5, and corollary 2.8, allow us to construct the most generic steady state of system (2.1), that is, the nullspace of $A$. From theorem 2.5, it follows that the set of non-trivial SCCs is exactly the set of final critical blocks, as defined before theorem 2.3. Hence $I_f \subseteq I_c$ is also the index set of non-trivial critical blocks, that is, $I_f = \{1 \leq k \leq h \mid B_k \text{ critical and non-trivial}\}$. All in all, a steady-state vector

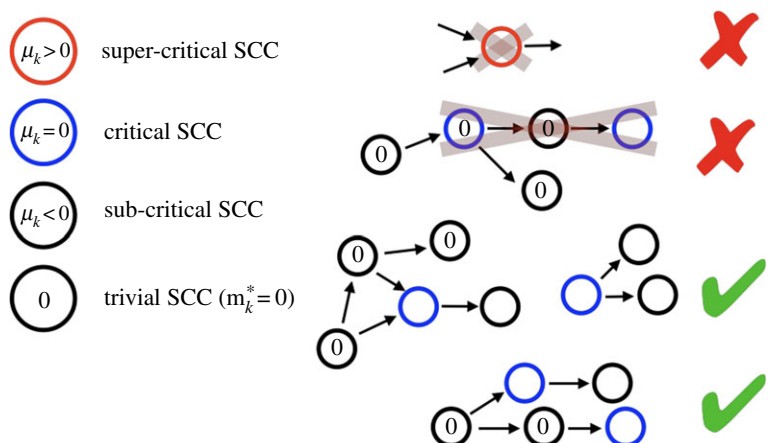

**Figure 2.** Illustration of theorem 2.9. Circles are SCCs, according to the condensation mapping as illustrated in figure 1 and coloured according to their type. For a linear cooperative system to be stable, all SCCs must have non-positive eigenvalues (no super-critical SCCs), and any SCCs with dominant eigenvalue zero (critical SCCs) cannot be connected by any directed path. Configurations which allow marginally stable states are shown with a green tick, and those which are unstable with a red cross. For the former, we also mark the trivial blocks. All non-negative, marginally stable states can be determined by setting zero all the nodes in all the trivial blocks, choose one 0-eigenvector for each critical (blue) block, and propagate them downstream using equations (2.8) and (2.9).

$\mathbf{m}^* = (\mathbf{m}_1, \mathbf{m}_2, \ldots, \mathbf{m}_h)^T$ has the form

$$\mathbf{m}_k^* = \begin{cases} 0 & \text{if } B_k \text{ is upstream of any critical block,} \\ \alpha_k \phi_k & \text{if } k \in I_f, \\ B_k^{-1}\left[\sum_{l \in I_f} P_{kl} \alpha_l \phi_l\right] & \text{if } B_k \text{ is sub-critical.} \end{cases} \tag{2.11}$$

This can also be written as

$$\mathbf{m}^* = \sum_{k \in I_f} \alpha_k \mathbf{m}_l^{*(k)}, \quad \text{with } \mathbf{m}_l^{*(k)} = \begin{cases} 0 & \text{if } B_l \text{ is upstream of } B_k, \\ 0 & \text{if } l \in I_f \text{ and } l \neq k, \\ \phi_k & \text{for } l \in I_f, \\ B_l^{-1}[P_{lk}\phi_k] & \text{if } B_l \text{ is sub-critical.} \end{cases} \tag{2.12}$$

Hence, the dimension of the nullspace of $A$ (equivalently, the geometric multiplicity of the eigenvalue zero) is equal to the number of non-trivial critical blocks. The algebraic multiplicity, on the other hand, is the number of all critical blocks, since according to the Perron–Frobenius theorem for Metzler matrices, each critical block has a simple (algebraic multiplicity 1) eigenvalue zero and thus contributes once to the multiset of eigenvalues of $A$, by equation (2.4). Recall that system (2.1) is marginally stable if and only if the dominant eigenvalue of $A$ is zero and its geometric multiplicity is equal to the algebraic multiplicity; this is thus the case only if there are no super-critical blocks and all critical blocks are non-trivial. According to theorem 2.5, this is the case if no critical block is upstream of another critical block, or alternatively, if there are no paths between any two critical blocks. We thereby arrive at

**Theorem 2.9.** *A dynamical system in the form of equation (2.1) with Jacobian matrix $A$ is marginally stable if, and only if, these conditions hold:*

 (a) *There are no super-critical blocks.*
 (b) *There is at least one critical block.*
 (c) *There are no (directed) paths in $G(A)$ which connect two critical blocks.*

Parts (a) and (b) follow from lemmas 2.1 and 2.2, while part (c) follows from equation (2.12) and theorem 2.5, that if there is a path between two critical blocks, one of them must be trivial. This theorem is illustrated in figure 2. To summarize our findings, including the necessary definitions, we can express the stability criteria of cooperative dynamical systems as follows:

**Theorem 2.10.** *Let $A = [a_{ij}]$, with $a_{ij} \geq 0$ if $i \neq j$, be the Jacobian matrix of the linear cooperative system in equation (2.1), and $G(A)$ its weighted graph, defined by the edge weights $a_{ij}$ for all $i$, $j$. Let $B_1, \ldots, B_h$ be the adjacency matrices of the strongly connected components (SCCs) of $G(A)$. We define an SCC as critical if its*

*dominant eigenvalue is zero, sub-critical, if its dominant eigenvalue is negative, and super-critical if its dominant eigenvalue is positive. Then*:

1. *The system is asymptotically stable if and only if all SCCs are sub-critical. In that case, the steady state vanishes (is the zero-vector).*
2. *Otherwise, the system is marginally stable if*
   (a) *there are no super-critical SCCs, and*
   (b) *there are no paths in G(A) which connect two critical SCCs.*
3. *Otherwise, the system is unstable.*

*The corresponding equilibrium set of the system is given by equation (2.11).*

## 3. Conclusion

The conditions stated in theorem 2.10 prescribe a way to simplify the analysis of a high-dimensional linear cooperative system by decomposition into lower dimensional subsystems, the strongly connected components (SCCs) of the dynamical system's dependence graph. By spectral analysis of these SCCs and checking whether the topological conditions of theorem 2.10 are fulfilled, the system's stability can be determined. In particular, *marginal stability* is of importance for linear systems, since marginally stable states represent the only possible non-vanishing stable states—i.e. not identical to the zero-vector—called *steady states*. Such a steady state features conservation of the quantity of interest 'on average', i.e. the mean value stays constant even if the quantity itself is not strictly conserved.[2] By contrast, asymptotically stable states are trivially vanishing for linear systems.

Moreover, our analysis revealed that a critical SCC, i.e. one with dominant eigenvalue zero, uniquely determines the steady state of the (necessarily sub-critical) SCCs upstream and downstream of it. In particular, the steady-state configurations of all SCCs downstream of a critical SCC do generally not vanish (theorem 2.3), and are uniquely determined by equations (2.8) and (2.9), while the steady-state configurations of all SCCs upstream of it must vanish (theorem 2.5). This leads to an explicit formula (equations (2.11) and (2.12)) to construct the steady state of the whole system by the knowledge of the steady states on the critical SCCs only.

The results, theorems 2.3, 2.5, 2.9 and 2.10, can be seen as a generalization of a similar condition found for linear *compartmental systems*, i.e. linear cooperative systems where the quantity of interest is strictly conserved (apart from external sources and sinks) [2]. For those systems, it has been found that the existence of at least one singular SCC (having an eigenvalue zero) is sufficient to ensure a non-trivial steady state [14,15]. Notably, due to the conservation law in compartmental systems, no SCCs with positive eigenvalue may exist, meaning that all singular SCCs are critical, and furthermore, no critical SCCs can have any outgoing links. Hence, the existence of a singular SCC in a compartmental system automatically implies the conditions of our theorem 2.9. The stability conditions of theorems 2.9 and 2.10 therefore represent a generalization to cooperative systems where the quantity of interest is not necessarily conserved, and can therefore also be applied to population dynamics where individuals can replicate and transit between different states. An example are populations of stem cells in animal tissues which differentiate, thereby changing their cell type. Note that while (cell) populations as a whole are often subject to feedback and thus follow nonlinear dynamics, when considering sub-populations therein, which compete neutrally, the corresponding subsystem is linear.

In general, cooperative systems can be highly complex, with a large number of variables and very complex interactions, hence represented by large and often irregular graphs. The method presented here is a way to significantly simplify the analysis of a wide range of systems, ranging from cooperative (bio-)chemical reactions to complex population dynamics, by decomposing the systems into their strongly connected components. We have shown that a spectral analysis of each SCC, and a simple graphical criterion of the connectivity between SCCs (theorem 2.10, figure 2) completely determine the stability of any linear cooperative system. This provides a unique insight into the possible configurations of cooperative systems and demonstrates the power of graph-theoretic techniques in the analysis of complex dynamical systems.

---

[2]We note that since a marginal steady state $m^*$ is a right 0-eigenvector, by fulfilling $Am^* = 0$, there must also exist a left 0-eigenvector $c$, fulfilling $cA = 0$. The latter equation defines a generalized conservation law with coefficients $c$; however, this conservation law may be non-trivial and cannot be directly derived from $m^*$.

Data accessibility. This article has no additional data.

Authors' contributions. P.G. and R.J.S.-G. carried out the mathematical analysis, P.G., B.D.M. and C.P. conceived and designed the project. All authors helped draft the manuscript. All authors gave final approval for publication.

Competing interests. We declare we have no competing interest.

Funding. P.G. was supported by a Medical Research Council New Investigator Research (grant no. MR/R026610/1). C.P. was supported by a Studentship of the Institute of Life Sciences.

Acknowledgements. We thank David Chillingworth for hinting us towards some literature on cooperative systems.

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
