## [Reviewer comments · Royal Society Open Science]

Review History

RSOS-191090.R0 (Original submission)

Review form: Reviewer 1

Is the manuscript scientifically sound in its present form?

No

Are the interpretations and conclusions justified by the results?

Yes

Is the language acceptable?

Yes

Do you have any ethical concerns with this paper?

No

Have you any concerns about statistical analyses in this paper?

No

Recommendation?

Major revision is needed (please make suggestions in comments)

Comments to the Author(s)

See report (Appendix A).

Review form: Reviewer 2

Is the manuscript scientifically sound in its present form?

Yes

Are the interpretations and conclusions justified by the results?

Yes

Is the language acceptable?

Yes

Do you have any ethical concerns with this paper?

No

Have you any concerns about statistical analyses in this paper?

No

Recommendation?

Reject

Comments to the Author(s)

In this article the authors find conditions for the stability of linear cooperative systems, based on graphical criteria of the underlying dependence graph. They are primarily interested in the case of non-trivial equilibria of the linear system which are Lyapunov stable, also referred to as marginally stable. This is the case if the eigenvalue of largest real part is zero and its geometric multiplicity is equal to its algebraic multiplicity. They show how marginal stability can be determined by decomposing the system into its strongly connected components (SCCs). The stability can then be inferred from (i) the spectrum of the Jacobian matrix of isolated SCCs, and (ii) the hierarchical arrangement of the SCCs (normal form of reducible matrix). The main result states that for (marginal) stability to prevail, no SCC may have positive eigenvalues, and any SCCs with eigenvalue zero may not stand in any hierarchical relation to each other, i.e. there may be no (directed) path connecting them.

The results of the paper are interesting and appear to be useful. My only concern is the extent to which they are novel. This concern is amplified by the paucity of the references included in the paper. Stability for compartmental systems is of great interest for compartmental modeling and Markov processes and so the literature must be vast and fragmented by discipline (probability, control theory, compartmental modeling, etc.).

In addition, the authors do not connect their main result with any related results in the literature. It is hard for this reviewer to believe that there are no such connections/comparisons.

As a consequence, this reviewer is reluctant to recommend publication because the authors fail to relate their result to those in the literature.

Some sources not included in the references known to this reviewer. Sources from probability theory are not included here:

J. Jacquez and C. Simon, Qualitative Theory of Compartmental Systems, SIAM Review 35 (1993) no.1 , pp 43-79.

G. Walter and M. Contreras, Compartmental Modeling with Networks, Birkhauser, 1999, Boston

Berman and Plemmons, Nonnegative Matrices in the Mathematical Sciences, SIAM Publications. ISBN: 978-0-89871-321-3

A. Berman, M. Neumann and R. Stern, Nonnegative matrices in dynamic systems, John Wiley & Sons, New York (1989)

Decision letter (RSOS-191090.R0)

23-Aug-2019

Dear Dr Greulich,

The editors assigned to your paper ("Stability and steady state of complex cooperative systems: a diakoptic approach") have now received comments from reviewers. We would like you to revise your paper in accordance with the referee and Associate Editor suggestions which can be found below (not including confidential reports to the Editor). Please note this decision does not guarantee eventual acceptance.

Please submit a copy of your revised paper before 15-Sep-2019. Please note that the revision deadline will expire at 00.00am on this date. If we do not hear from you within this time then it will be assumed that the paper has been withdrawn. In exceptional circumstances, extensions may be possible if agreed with the Editorial Office in advance. We do not allow multiple rounds of revision so we urge you to make every effort to fully address all of the comments at this stage. If deemed necessary by the Editors, your manuscript will be sent back to one or more of the original reviewers for assessment. If the original reviewers are not available, we may invite new reviewers.

- Data accessibility

<http://datadryad.org/submit?journalID=RSOS&manu=RSOS-191090>

- Competing interests

- Authors' contributions

- Acknowledgements

- Funding statement

on behalf of Dr James Locke (Associate Editor) and Mark Chaplain (Subject Editor)
openscience@royalsociety.org

Associate Editor's comments (Dr James Locke):

Although both reviewers enjoyed aspects of the work, there were some major concerns over novelty from both reviewers. This should be addressed in a major revision of the paper, with the appropriate references discussed and the new aspects of the work clearly explained.

Reviewers' Comments to Author:

Reviewer: 1

See report.

Reviewer: 2

Comments to the Author(s)

In this article the authors find conditions for the stability of linear cooperative systems, based on graphical criteria of the underlying dependence graph. They are primarily interested in the case of non-trivial equilibria of the linear system which are Lyapunov stable, also referred to as marginally stable. This is the case if the eigenvalue of largest real part is zero and its geometric multiplicity is equal to its algebraic multiplicity. They show how marginal stability can be determined by decomposing the system into its strongly connected components (SCCs). The stability can then be inferred from (i) the spectrum of the Jacobian matrix of isolated SCCs, and (ii) the hierarchical arrangement of the SCCs (normal form of reducible matrix). The main result states that for (marginal) stability to prevail, no SCC may have positive eigenvalues, and any SCCs with eigenvalue zero may not stand in any hierarchical relation to each other, i.e. there may be no (directed) path connecting them.

The results of the paper are interesting and appear to be useful. My only concern is the extent to which they are novel. This concern is amplified by the paucity of the references included in the paper. Stability for compartmental systems is of great interest for compartmental modeling and Markov processes and so the literature must be vast and fragmented by discipline (probability, control theory, compartmental modeling, etc.). In addition, the authors do not connect their main result with any related results in the literature. It is hard for this reviewer to believe that there are no such connections/comparisons.

As a consequence, this reviewer is reluctant to recommend publication because the authors fail to relate their result to those in the literature.

Some sources not included in the references known to this reviewer. Sources from probability theory are not included here:

J. Jacquez and C. Simon, *Qualitative Theory of Compartmental Systems*, SIAM Review 35 (1993) no.1 , pp 43-79.

G. Walter and M. Contreras, *Compartmental Modeling with Networks*, Birkhauser, 1999, Boston

Berman and Plemmons, *Nonnegative Matrices in the Mathematical Sciences*, SIAM Publications. ISBN: 978-0-89871-321-3

A. Berman, M. Neumann and R. Stern, *Nonnegative matrices in dynamic systems*, John Wiley & Sons, New York (1989)

Author's Response to Decision Letter for (RSOS-191090.R0)

See Appendix B.

Decision letter (RSOS-191090.R1)

04-Nov-2019

Dear Dr Greulich,

I am pleased to inform you that your manuscript entitled "Stability and steady state of complex cooperative systems: a diakoptic approach" is now accepted for publication in Royal Society Open Science.

on behalf of Dr James Locke (Associate Editor) and Mark Chaplain (Subject Editor)
openscience@royalsociety.org

Associate Editor Comments to Author (Dr James Locke):

Thank you for your careful revisions of the manuscript. The manuscript is now acceptable for publication.

Appendix A

Review of “Stability and steady state of complex cooperative systems: a diakoptic approach”

July 16, 2019

1 General comments

I enjoyed reading this article. It provides a simple characterisation of marginal stability of a linear dynamical system, derived from a ‘coarse-grained’ picture of the dynamics expressed through the strongly-connected components of the associated graph.

I cannot vouch for the originality of the results as I am not an expert in the theory of positive linear systems. I will make some suggestions for extra referees.

From an applications side, I think it would improve the future impact of the paper to have a clear example where calculation of marginal stability is important and useful. The short stochastic processes discussion in section 3 is nice for context, but the results are not new. Relatedly, it would be nice to have some clear discussion on the relationship between marginal stability and conservation laws.

2 Corrections

2.1 Major

I believe there is an error in the proof of Theorem 2, around lines 34–36. The authors state correctly that $B_k[e^{B_k t} \mathbf{m}_k^*] \rightarrow 0$. This is true because $e^{B_k t}$ converges to a projection matrix, whose range is spanned by the dominant eigenvector of B_k . From (6), it therefore follows that $\lim_{t \rightarrow \infty} e^{B_k t} [\sum_{l < k} C_{kl} \mathbf{m}_l^*] = 0$, however, this does not imply $\sum_{l < k} C_{kl} \mathbf{m}_l^* = 0$ in general as claimed. All it gives you is that $\sum_{l < k} C_{kl} \mathbf{m}_l^*$ is in the kernel of the projection $\lim_{t \rightarrow \infty} e^{B_k t}$. (Even though $e^{B_k t}$ is invertible for all t , the limiting projection is not invertible.)

To fix the gap in the proof you can use the non-negativity of C and \mathbf{m}^* . If any non-negative vector has a non-zero entry then it cannot be orthogonal \mathbf{m}_k^* , which is known to be positive, hence $\sum_{l < k} C_{kl} \mathbf{m}_l^*$ must be empty.

2.2 Minor

p2 l16 “studied isolated” should be “studied in isolation”

p2 l48 I don’t think the SIR example here is a good one, since there is a non-linear interaction between S and I, and the Jacobian contains negative off-diagonal terms

p3 l34 I think the possessive form of “process” is “process’s”

p4 l17 Why does x turn to m here?

p5 l13 “graphs” should be “graph’s”

p9 l13 The meaning of “final” is not clear here

Appendix B

Philip Greulich

University Lecturer

University of Southampton

UNIVERSITY OF
Southampton

School of Mathematics
Faculty for Social Sciences

Re: Revised Manuscript: “Stability and steady state of complex cooperative systems: a diakoptic approach ”
by Philip Greulich, Ben MacArthur, Cristina Parigini,
and Ruben Sanchez-Garcia

Dear Editor,

Please find enclosed a revised version of the manuscript RSOS-191090 “Stability and steady state of complex cooperative systems: a diakoptic approach” and a detailed response to the reviewers’ remarks further below.

We are pleased that both reviewers find our manuscript interesting and useful for the community. Reviewer 1 is very positive in his assessment and has some suggestions for improvement, which we implement in the revised version. We explicitly thank the reviewer for pointing out a gap in one proof, which we have now corrected in the revised version (see also response below) and for which the reviewer’s comments were very helpful.

Reviewer 2 had some concerns about the novelty with regards to compartmental systems, a subclass of cooperative systems which are characterised by an additional conservation law for the studied quantities. We agree with the reviewer that for this subclass of cooperative systems similar approaches based on graph-theoretical arguments are already known. We have now clarified this in the revised manuscript and extended our literature review accordingly. Crucially, however, our results go well beyond compartmental systems, and are applicable more generally also to non-conserved linear cooperative systems, which may include replication of the considered quantity. In particular, our results also apply to the wide field of population dynamics, such as populations of individuals in different life cycle stages, or tissue cell populations which proliferate and differentiate (e.g. stem cells). We note that the previously known results for compartmental dynamics are not applicable to those biologically and medically important systems, and our analysis is fundamentally different to approaches for compartmental systems.

To our knowledge and given evidence provided by Reviewer 2, as well as from an extended literature review, we are therefore convinced that our results are entirely novel, with respect to linear cooperative systems in general, that include population dynamics. We thus believe that our work provides a valuable and simple-to-use tool to study the stability of replicating populations which is of crucial importance in ecology, developmental biology, and biomedicine.

With kind regards,

Philip Greulich, Ben MacArthur, Cristina Parigini, and Ruben Sanchez-Garcia

Detailed response to the reviewers’ remarks:

Reviewer: 1

COMMENTS TO THE AUTHOR(S)

I enjoyed reading this article. It provides a simple characterisation of marginal stability of a linear dynamical system, derived from a ‘coarse-grained’ picture of the dynamics expressed through the strongly connected components of the associated graph.

We are pleased about the reviewer's positive view of our manuscript. Below we have addressed the reviewer's remarks.

From an applications side, I think it would improve the future impact of the paper to have a clear example where calculation of marginal stability is important and useful?

We have now expanded on the real world applications of our results both in the Introduction and the Discussion section. We now describe more explicitly the application to population dynamics and in particular to tissue cell populations which can proliferate (stem and progenitor cells) and change their state through differentiation. Population dynamics are linear when considering neutrally competing (sub-)populations. We note that in a linear system, marginal stability is the only possibility to maintain a non-zero cell population, which corresponds to a healthy tissue cell population. In contrast, the only possible asymptotically stable state in a linear system is when a population vanishes, but is not a biologically viable state. The biomedical importance to identify a marginally stable – i.e. healthy – state of a tissue, is also with respect to the potential transition to cancer, which is related to an unstable tissue cell population.

The short stochastic processes discussion in section 3 is nice for context, but the results are not new.

We agree with the reviewer and have now removed this section, to avoid the misleading impression that these results were novel (see also response to Reviewer 2).

Relatedly, it would be nice to have some clear discussion on the relationship between marginal stability and conservation laws.

Only for compartmental systems, the quantity of interest is conserved. For cooperative systems in general, marginal stability does only mean conservation "on average", i.e. that the mean value stays constant over time. We note, however, that since the marginally stable is a right 0-eigenvector to the matrix A (eigenvector to eigenvalue zero), there must also be a left 0-eigenvector to A , and this left 0-eigenvector marks the coefficients of a non-trivial conservation law, meaning that a linear combination of the "masses" on each node exists that is conserved (the coefficients of this linear combination are the entries of the left 0-eigenvector). If these coefficients were all '1's, then this would correspond to standard conservation of the quantity of interest, but this is in general not the case. Furthermore, the corresponding left eigenvector (i.e. the conservation law) can, to our knowledge, not be easily derived from the knowledge of the right eigenvector (i.e. the marginally stable state). We have commented on that in the revised Conclusions section, where this relationship is clarified in a footnote. However, we do not expand on this analysis as this would go beyond the scope of the manuscript.

I believe there is an error in the proof of Theorem 2, around lines 34{36. [...]

We sincerely thank the reviewer for spotting this gap in our proof. Indeed, this part of the proof was not entirely correct, and we have revised this part of the manuscript to now present a correct, consistent proof in the text. Part of this proof is now separated as Lemma 4. We apologise for not having spotted this gap in the proof before. In particular, we wish to thank the reviewer for his suggestions how to solve this issue.

"Minor:"

We have corrected all minor issues pointed out by the reviewer. In particular, we have now removed any reference to the SIR model and epidemic models in general, as they are not good examples.

Reviewer: 2

The results of the paper are interesting and appear to be useful. My only concern is the extent to which they are novel. This concern is amplified by the paucity of the references included in the paper. Stability for compartmental systems is of great interest for compartmental modeling and Markov processes and so the literature must be vast and fragmented by discipline (probability, control theory, compartmental modeling, etc.). [...]

Some sources not included in the references known to this reviewer. Sources from probability theory are not included here:

J. Jacquez and C. Simon, *Qualitative Theory of Compartmental Systems*, SIAM Review 35 (1993) no.1 , pp 43-79.

G. Walter and M. Contreras, *Compartmental Modeling with Networks*, Birkhauser, 1999, Boston

Berman and Plemmons, *Nonnegative Matrices in the Mathematical Sciences*, SIAM Publications. ISBN: 978-0-89871-321-3

A. Berman, M. Neumann and R. Stern, *Nonnegative matrices in dynamic systems*, John Wiley & Sons, New York (1989)

We thank the reviewer for hinting at further literature sources, in particular with respect to compartmental systems, a sub-class of cooperative systems which is characterized by a *conserved quantity* (in the sense that the quantity of interest cannot replicate – gain and loss through sources and sinks is possible also in compartmental systems). We have now expanded our literature review (in the introduction) to include those references and relevant sources cited therein. We also performed an extensive further literature review, whose outcome is now reflected in the manuscript. We note, however, that neither the references provided by the reviewer, nor other references found through this literature research, provide graph-based (diakoptic) criteria for marginal stability in generic cooperative systems, when also non-conserved quantities are considered (for example populations of replicators).

We therefore conclude that to our knowledge, and including the evidence given by the reviewer as well as from our extensive literature review, our findings are entirely novel with regards to cooperative systems where the studied quantity is not conserved.

Nonetheless, we agree with the reviewer that a similar approach, based on identifying strongly connected components of the system, has been done for (conserved) compartmental systems, which include Markov processes. We therefore apologise if we created the impression to claim novelty for those systems. In order to clarify this, we have substantially rewritten the Introduction and Conclusions sections, to discuss the relevant known results for compartmental systems in both sections. To clarify the novel aspect of our work, we now emphasize the applicability of our study to cooperative systems without a conserved quantity, which are encountered in the real world for example in population dynamics: populations of individuals and cells can replicate and can therefore not be described by standard compartmental systems. We further clarify a misleading statement in the beginning of the results section (on page 4 before Eq. (1)), that possibly created the impression that we consider only mass ratios. Finally, since Markov processes are conserved systems, we removed the section on steady states of Markov processes, in order to avoid the impression that these are novel results, since existing methods can be applied to study them.

In addition, the authors do not connect their main result with any related results in the literature. It is hard for this reviewer to believe that there are no such connections/comparisons.

In the revised manuscript, we now discuss, both in the Introduction and the Conclusions, our results in comparison to what is known about general cooperative systems (in particular the results of Hirsch and Smith) and relate those to similar approaches for compartmental systems.